# Weathering of Antibacterial Melt-Spun Polyfilaments Modified by Pine Rosin

**DOI:** 10.3390/molecules26040876

**Published:** 2021-02-07

**Authors:** Mikko Kanerva, Jacob Mensah-Attipoe, Arja Puolakka, Timo M. Takala, Marko Hyttinen, Rama Layek, Sarianna Palola, Vladimir Yudin, Pertti Pasanen, Per Saris

**Affiliations:** 1Faculty of Engineering and Natural Sciences, Tampere University, FI-33014 Tampere, Finland; ao.puolakka@gmail.com (A.P.); rama.layek@lut.fi (R.L.); sarianna.palola@tuni.fi (S.P.); 2Department of Environmental and Biological Sciences, University of Eastern Finland, P.O. Box 1627, FI-70211 Kuopio, Finland; jacob.mensah-attipoe@uef.fi (J.M.-A.); marko.hyttinen@uef.fi (M.H.); pertti.pasanen@uef.fi (P.P.); 3Department of Microbiology, Helsinki University, P.O. Box 56, FI-00014 Helsinki, Finland; timo.takala@helsinki.fi (T.M.T.); per.saris@helsinki.fi (P.S.); 4Department of Separation Science, School of Engineering Science, LUT University, Mukkulankatu 19, FI-15210 Lahti, Finland; 5Institute of Macromolecular Compounds, Russian Academy of Sciences, 31 Bolshoy pr. VO, 199004 Saint-Petersburg, Russia; yudinve@gmail.com

**Keywords:** durability, antibacterial response, pine rosin, volatile organic compounds

## Abstract

For many antibacterial polymer fibres, especially for those with natural functional additives, the antibacterial response might not last over time. Moreover, the mechanical performance of polymeric fibres degrades significantly during the intended operation, such as usage in textile and industrial filter applications. The degradation process and overall ageing can lead to emitted volatile organic compounds (VOCs). This work focused on the usage of pine rosin as natural antibacterial chemical and analysed the weathering of melt-spun polyethylene (PE) and poly lactic acid (PLA) polyfilaments. A selected copolymer surfactant, as an additional chemical, was studied to better integrate rosin with the molecular structure of the plastics. The results reveal that a high 20 w-% of rosin content can be obtained by surfactant addition in non-oriented PE and PLA melt-spun polyfilaments. According to the VOC analysis, interestingly, the total emissions from the melt-spun PE and PLA fibres were lower for rosin-modified (10 w-%) fibres and when analysed below 60 °C. The PE fibres of the polyfilaments were found to be clearly more durable in terms of the entire weathering study, i.e., five weeks of ultraviolet radiation, thermal ageing and standard washing. The antibacterial response against Gram-positive *Staphylococcus aureus* by the rosin-containing fibres was determined to be at the same level (decrease of 3–5 logs cfu/mL) as when using 1.0 w-% of commercial silver-containing antimicrobial. For the PE polyfilaments with rosin (10 w-%), full killing response (decrease of 3–5 logs cfu/mL) remained after four weeks of accelerated ageing at 60 °C.

## 1. Introduction

Sustainability will be a crucial property in the development of new materials. Due to the necessities of circular economy and sustainable balance between us and our alive surroundings, large amounts of bio-based polymers and polymers with natural, degradable components have been introduced in the literature during the recent years [1,2,3]. These materials are not of practical use, if the intermediate forms, like fibres and polyfilaments, can not reach the performance requirements or if the safety towards living organisms is unacceptable in the operational environment.

Fibres, having two of the three dimensions small, are prone to affect their environment because of the high surface to volume ratio. Medicinal and other fibres with an antimicrobial response are developed to protect against virus and bacterial strains that are hazard to a human [4,5,6]. Silver is typical additive to make antimicrobial fibres [7,8,9]. Silver itself is considered an expensive and toxic element and related modifications for antimicrobial compounds, i.e., polymer systems including nano particles and surface functionalization [10], rarely improve this character [11,12]. Natural components could be a solution for healthier, profitable and sustainable additives in functional antimicrobial fibres.

In spite of the development work and scientific characterization of numerous reported antimicrobial fibres and polymers, the detailed understanding of the long-term durability and related degradation is not well understood for functional polymer fibres with natural additives. Typically, the durability of the antimicrobial response is much less than the mechanical durability of the basis polymer structure [13,14]. The durability at elevated temperatures is especially important because the degradation products might get emitted in a gaseous form during the use of a textile product; some degradation can occur already during processing after the fibre cool-down [15]. The durability of polymer fibres with natural components at elevated temperatures and/or moist environment is typically lower compared to synthetic rivals due to the (natural) additives [16].

Pine rosin has been researched in various antimicrobial polymer systems for a few decades and its antibacterial response has led to many applications related to medical wound healing, for instance [17,18]. The chemical content of rosin (resin) depends naturally on the exact tree species, the part of tree being used and the extraction process [19,20,21]. Rosin alone or its derivatives as well rosin as additive in various polymeric systems have been proven antimicrobial against several Gram-positive and Gram-negative bacteria [22,23,24] and fungi [25,26,27]. Even when rosin is externally subject to living organisms, it can form low-molecular weight emissions (volatile organic compounds, VOCs) that could be defined as harmful in large quantities [28,29]. Moreover, the higher the temperature and longer the exposure time, the stronger the released emission and degradation of rosin are expected to be. However, in-depth studies about the mechanical durability, VOC emissions, along with antibacterial response in advanced polymer systems, such as melt-spun polyfilaments, have not been reported.

## 2. Background

Rosin in polymeric fibres and textiles has great potential to bring in multiple benefits for the final products. First, rosin-polymer polyfilaments and the formed textiles or filters have a strong antibacterial response. Second, rosin (or tall oil) itself is a by-product of the forest industry [30]—the use of rosin increases the sustainable efficiency of forest industry and rosin can be a cost-efficient additive for antimicrobial products. Third, rosin is a completely natural additive, thus, the higher the content of rosin in the final product, the higher is the bio content of the product. In the current literature, high rosin concentrations have been studied for several polymers: polyamide 6 (PA), polyethylene (PE), polypropylene, poly lactic acid (PLA) and a starch-based polymer compound [6]. The reported work focused on the polyfilament melt-spinning of these polymers and blends at 0…20 w-% concentration of rosin. The maximum rosin concentration was determined to be ≈10 w-% [6]. A high (≈20 w-%) rosin dosage leads to disintegration and accumulation of rosin at the spinneret and subsequent unstable flow and fibre formation during polyfilament spinning; the melt viscosity of rosin and its low-molecular weight components are relatively low.

The reported problems at high rosin concentrations might have been due to the low compatibility and incomplete mixing of rosin within the polymer melt. For this, a surfactant chemical could be used to improve the melt compatibility. Typically, the surfactant’s type is selected based on the surface functionality of the compounds to be mixed. Rosin is the mixture of a wide spectrum of compounds including phenolic compounds, waxes, a large variety of fatty acids as well as terpenoid, sterol and terpene (resin) acids. In turn, polyolefins do not have special functionality. Since ionic surfactant is typically preferred for an aqueous dispersion and solid particles [31,32], polymeric surfactant is presumed to be a suitable candidate for rosin-polymer blends. The improvements by the selected surfactant could be listed as enhanced spinning process and quasi-static mechanical properties (stiffness, ultimate strength, ductility). For the following research, a non-ionic copolymer surfactant was selected. The proper surfactant concentrations for dispersing plant-based natural materials have been reported to be 0.09…0.3 w-% (of dry weight) in the current literature [33]. In the following research, a concentration of 0.166 w-% (of the total dry weight, 1% of rosin weight) is selected to be used with the polymer-rosin blends.

In the work by Kanerva et al., the functionality, i.e., the antibacterial response against Gram-positive *Staphylococcus aureus* and Gram-negative *Escherichia coli*, was reported to be highly dependent on the type of polymer applied. The strongest reported response was achieved with rosin-containing polyfilaments with either PA, PE or PLA basis [6]. However, no comparison with any rival synthetic additive was made in terms of the antibacterial response. To understand the level of antibacterial response of the rosin-containing polyfilaments, compared to a synthetic commercial additive, a selected silver-containing additive is studied in the following research. The concentration range (1–2 w-%) of the silver-based additive is selected based on the existing literature (2.0 w-%) [9]. An initial survey indicated similar performance for 1 w-% and 2 w-% additive concentrations. Therefore, the antibacterial response of the polyfilament series with the lower (1 w-%) dosage is reported as a realistic (economical) comparison with rosin.

The long-term durability of the antibacterial response was not reported in the work by Kanerva et al. [6]. For textile and filter applications, for example, fibres are exposed to ultraviolet (UV) radiation from the sun and they degrade in terms of mechanical properties. Various clothing and filter applications are possible for melt-spun polyfilaments. For standard washing programs, the highest temperatures can be detrimental to a polymeric product. Weathering, including thermal cycling, washing and ultraviolet (UV) irradiation, is considered for the PE and PLA polyfilaments with a rosin concentration of 10 w-% in the following research.

## 3. Materials and Methods

### 3.1. Polymer Raw Materials and Fibre Melt-Spinning

The polymer raw materials used for the compounding and fibre spinning in this work are given in Table 1. As-received granulate was dried before compounding. To achieve antibacterial response, pine gum rosin (acid value 167 mg KOH/g, softening point 74 °C) (Ro) by Forchem (Finland) was used as natural functional additive; the spinning performance of rosin in various melt-spun polyfilaments has been recently reported [6]. The characterization of the pine rosin grade used can be found in previous works [6,34]. All of the polymer blends of this study were compounded by using a model TSE 25 twin-screw extruder (Brabender, Germany). After extruding a string of the compounded polymer system, it was cut into 2–4 mm-size granulate particles by a mechanical crusher. The granulate per polymer was used to feed the polyfilament spinning apparatus. All of the polyfilaments in this study were melt-spun by using a polyfilament spinning system (Fourné Polymertechnik GmbH, Germany) and more information about the spinning process can be found in a recent work [6]. No lubricant or fibre finish was used during the spinning process. The main processing parameters of the blends of this study are given in Table 2. It should be noted that no orientation was applied for the polyfilaments of this study.

Due to the challenges followed by very high rosin concentrations, bio-reagent type copolymer surfactant (PF) F-127 by Pluronic, purchased from Sigma Aldrich (Merck KGaA, Germany, Darmstadt) was studied here. The surfactant-containing blends were prepared as follows: Firstly, 300 g of rosin was mixed with surfactant (1 w-% of rosin dry weight) by manual grinding using a ceramic mortar pestle and, then, kept in an oven (100 °C). Secondly, the mixture was cooled to room temperature and crushed well. This procedure was repeated three times to obtain surfactant-modified rosin. The surfactant-modified polymer blends (see Table 3) were obtained by melt-compounding of the mixture of surfactant and rosin with as-received polymer granulate.

Commercial ‘synthetic’ medical-purpose silver nanoparticles (Poviargol, Russia) were used to compound and melt-spin and analyse a comparative polyfilament series. The silver-containing (Ag) compounds were prepared by mixing as-received polymer granulate with the Ag-additive (powder-like form). Then, the mixture was fed to a compounder in small quantities (≈50 g) with a masterbatch basis. The desired diluted blends were compounded immediately to have 1% (and 2% for an initial survey) Ag-additive concentrations per compound (see Table 4). The Ag-containing particles were characterized using scanning electron microscopy (SEM) by a device Zeiss ULTRAplus (Zeiss, Germany) and elemental analysis by an integrated sensor for X-ray energy dispersive spectroscopy (EDS) (INCA Energy 350 EDS analyser, INCAx-act detector, Oxford Instruments, Oxford, UK).

### 3.2. Antibacterial Activity

The antimicrobial activity of the various polymer systems was tested against indicator bacteria *Staphylococcus aureus* ATCC 12598. The indicator was cultured at 37 °C in lysogeny broth (LB), with 1.5% agar for solid media. Antibacterial tests with the material samples were carried out in Ringer’s solution of 1/4 strength (i.e., mixture of NaCl, KCl, ${\mathrm{CaCl}}_{2}$, ${\mathrm{NaHCO}}_{3}$ and distilled water). The sample fibres were collected so that a mass of 0.1–0.14 g was used per fibre series and, for granulate form, 0.5 g. The indicator strain was first cultured overnight in LB broth. Colony forming units per mL of the o/n culture was determined by serial dilutions in 1/4 strength Ringer’s solution and plating onto LB agar. From the serial dilutions of $10^{3}$–$10^{4}$, 1 mL (about $10^{5}$–$10^{7}$ cfu/mL) was used for the antimicrobial test by mixing with the sample fibres inside 2-mL Eppendorf tubes. The mixtures were incubated for 24 h at a room temperature in a rotator (at 22 rpm). After incubation, the samples were serially diluted in 1/4 strength Ringer’s solution, and plated onto LB agar for determining the bacterial survival by colony counting. At least duplicates were analysed. In this study, the control and reference samples were polyamide 6 polyfilaments with zero and 10 w-% rosin, respectively (series names fPA and fPA10), from the polyfilament fibre batch with details of preparation reported in a recent work [6].

### 3.3. Volatile Organic Compounds Analysis

Volatile organic compounds (VOC) were analysed for the polymer fibres to understand the safety and stability at elevated temperatures. Two parallel sets of seven fibre samples (of which five reported here, fPE, fPE10, fPLA, fPLA10 and fPLA1Ag) were analysed. The polyfilament fibres (≈1 g) were wrapped around stainless steel cylinders (cylinders acetone-washed and dried, individual cylinder per fibre sample used) to form neat bundles of fibre. The bundles were removed from the cylinders prior to the analysis into the measurement chambers. For the analysis, the samples were placed in the chambers of M-CTE250 (M-CTE250, Markes Micro-chamber/Thermal Extractor, Markes International, UK); the chambers were acetone-washed and dried by heating in an oven at 180 °C for three hours. Before the analysis, each sample was weighed and analysed for emissions at three temperatures: (25 °C, 60 °C and 105 °C). The materials were heated with the M-CTE250. Before sampling, the materials were allowed to equilibrate at a set temperature for five minutes. The temperature of the MCTE250 chambers was ramped up from the lowest to the highest temperature per test. The chemicals emitted from the materials were collected into Tenax TA tubes at an average flow rate of 63 mL/min for five minutes for two first temperatures (25 °C and 60 °C) and for one minute when sampling at 105 °C.

At the end of a sampling interval, the tubes were removed and capped with the brass storage caps and, thereafter, analysed with the GC-MS. The Tenax TA tubes were analysed by a gas chromatograph (7890, Agilent, US) equipped with a mass selective detector (5975C, Agilent, Santa Clara, CA, USA) after thermal desorption (TD) (Markes TD-100, Markes International, Llantrisant, UK). The oven program included: 38 °C for 4 min then 5 °C/min to 210 °C for 0 min and finally 20 °C/min to 280 °C for 6.5 min. The total run time for a sample was 48.4 min. The transfer line temperature was 350 °C and it was directly connected to HP-5 silica capillary column (60 m × 250 µm × 0.25 µm). The resolved analytes were detected using an electron impact mass spectrometer (MS) (5975, Agilent Technologies, Santa Clara, CA, USA). The identification of the compounds was accomplished by the retention times, standard compounds and GC-MS data library. The MS was operated in the scan mode. The standard tubes were analysed similarly as the sample tubes. The total volatile organic compounds (TVOCs) values were defined as the integrated detector response value in toluene equivalents of compounds eluting between and including C6 to C16 as given in the standard ISO 16000–6:2004. The detailed (tabulated compounds) VOC data and chromatograms (total ion chromatograms) per sample material are included as Appendix A.

### 3.4. Conditioning by UV Radiation

Outdoor degradation of the fibres was simulated by subjecting the polyfilaments (loose fibre loops were formed) to UV-B radiation (UVA-340 lamps, Q-Lab, Westlake, OH, USA) with peak intensity of 340 nm and dose of 0.001 GJ/($m^{2}$ h) inside a laboratory UV-cabinet (test area 0.89 $m^{2}$). The resistance against the radiation was analysed with a combination of tensile testing and conditioning in the UV-cabinet (temperature in the cabinet 30–31 °C, total exposure time 0–504 h). Single fibre samples were extracted for mechanical testing every seven days for tensile testing.

### 3.5. Conditioning by Thermal Cycling

Pure thermal effects where studied by a combination of tensile testing and thermal ageing in an oven. The fibres (loose loops were formed) were laid in a digitally controlled, air-circulating oven (60 °C). Single fibre samples were extracted from the loops every seven days for mechanical testing to understand the effects of elevated temperature.

### 3.6. Conditioning by Standard Washing

The durability against standard washing (e.g., related to anticipated textile/clothing applications) was analysed with a combination of tensile testing and standard washing cycles according to the standard SFS-EN ISO 6330. The standard washing machine (Wascator FOM71MP-Lab, Electrolux, Sweden) was used at a temperature of 70 °C, with 20 g of IEC-detergent per wash, and, ten subsequent washing cycles were performed. Each polyfilament yarn was loosely fixed to form loops, inside the machine.

### 3.7. Mechanical Testing of Fibres from Polyfilaments

Mechanical properties, i.e., ultimate (peak) engineering stress and ultimate strain (at break) were measured by using a tensile fibre tester (Favigraph, Textechno, Germany). For calculating the stress values (peak force per cross-sectional area), fibre (filament) diameters were measured individually per test fibre; the individual fibre diameter was measured by using a visual light (VL) microscope. The test rate was 20 mm/min and a pre-tension of ≈0.7 cN was applied. By default, ten parallel fibre samples of a polyfilament were measured per blend series and average and standard deviations were reported.

## 4. Results and Analysis

### 4.1. Performance of As-Spun Polyfilaments in the Room Conditions

The antibacterial response of PE and PA-6 fibres with 10 w-% of rosin, against *S. aureus* bacteria, was recently reported to be approximately four logs (input ≈300,000) during 24 h of mixing (Ringer’s solution) at a room temperature [6]. The reported response of granulate (polymer prior spinning) was even higher, almost five logs (input over one million). For PLA and 10 w-% of rosin, the response was significant but clearly less, one log of decrease. In this study, the as-spun polyfilament fibres (PA-6 polymer basis) of the same batch (as in a work [6]) repeatedly created the same response (≈4 logs) with the current bacterial input of over one million. It should be noted that the fibres without rosin do not have antibacterial response (within the experimental scatter). To compare the rosin-induced antibacterial response to that of the Ag-containing synthetic rival, the results are clear in Figure 1. Similarly to the fibre-granulate dependence in the previous work, the response of the Ag-containing granulate (prior to spinning) is higher than that of the melt-spun fibre form. This suggests that the silver-containing particles are not located on the melt-formed surface of fibre as much as inside the fibre bulk—contrary to the granulate that always includes the fracture surface (due to the crushing process) that exposes the inside of the polymer bulk to the bacteria. Alternatively, for rosin-containing fibres, it could indicate removal of part of rosin (vaporized emissions) from the polymer bulk during the melt-spinning. In general, the antibacterial response of the rosin-containing fibres (fPA10 series [6]) was at the same level than the response of the fibres (and granulate) with the 1 w-% concentration of Ag-additive.

SEM imaging indicated essentially spherical morphology for the silver-containing additive particles, shown in Figure 2. The EDS spectrum revealed approximately 58%-content of silver. The particles have been treated by poly(vinylpyrrolidone) by the manufacturer to form compatible surfaces and stable dispersion with various polymer systems [9]. Here, poly(vinylpyrrolidone) was indicated by the oxygen (nitrogen) content in the spectrum (Figure 2b). The high silver content of the additive agrees with the good antibacterial response of the Ag-containing polyfilaments of this study. The efficacy of nano-sized silver and poly(vinylpyrrolidone) alone against *S. aureus* has been reported in the current literature [10].

The tensile mechanical properties of the melt-spun polyfilaments in the room conditions are summarized in Table 5. Based on the revealed properties and spinnability, the use of surfactant with blends of 20 w-% rosin led to stable spinning performance and acceptable mechanical strength. However, the ultimate strength was 7–40% lower compared to pure PE fibres (spinning at 160 °C and 180 °C) and 38% lower compared to fibres with 10 w-% rosin content. For the PLA systems, the surfactant modification—likewise for PE—allowed spinning at a 20 w-% rosin content but the determined ultimate strength of the fibres was essentially the same as for the PLA fibres with a 10 w-% rosin content. Additionally, the fibres spun with the surfactant dosage were brittle (low strain at break). In summary, the use of the selected surfactant did not essentially improve the mechanical performance of rosin-containing PE and PLA fibres in the room conditions. Therefore, the PF-series fibres were not studied further for long-term durability. For an application, where merely a high rosin content is important in the final product, the use of surfactant could be considered.

For the Ag-containing PE and PLA fibre series, compounds were successfully prepared and polyfilament fibres melt-spun. The powder-form Ag-additive dispersed well within the polymer blend based on visual observation. Similarly, the results with a similar performance as for PE and PLA fibres without the Ag-additive indicate good mixing during compounding. In detail, the fPE1Ag series PE fibres had 13–16% lower strength compared to pure PE and 10 w-% of rosin containing fibres (spinning at 160 °C). The fPLA1Ag series PLA fibres had only 5% lower strength and 21% lower strain at break—the results suggested better compatibility of the Ag-additive with PLA than with the PE grade used. In summary, the Ag-additive at 1 w-% concentration resulted in appropriate melt-spun polyfilaments, and, it does represent a mechanically realistic synthetic reference for the research of antibacterial response of polyfilaments with rosin.

### 4.2. Stability of Polyfilaments in Terms of Volatile Organic Compounds at Elevated Temperatures

The melt-spun polyfilaments with PE and PLA basis were studied in terms of VOC to estimate the stability of the fibres at immediate elevated temperatures. The total volatile organic compounds (TVOCs) detected in the materials at 25 °C, 60 °C and 105 °C are presented in Table 6. It can be seen that PE fibres emit minor TVOCs already at the room temperature (25 °C). For PLA basis, the range of observed concentrations range from zero to a very low level through the measurement at the room temperature. Interestingly, for all the fibres, the rosin-containing fibres resulted in significantly (23–84%) lower emitted organic compounds (TVOCs). This observation suggests that rosin could form a film or skin on the outer surface of the melt-spun polyfilaments that prevents the organic compounds being emitted from inside the bulk. When considering the results of antibacterial response (comparison between fibre and granulate), this is an opposite indication. Therefore, the lower TVOCs might indicate that rosin in general stabilizes the polymer-rosin blend (at low temperatures). This trend continued for PE and PE-rosin fibres at 60 °C. On the contrary, the measurement of fPLA10 series resulted in 46% higher TVOCs, although, the range of measured concentration during the measurement was broad (range 81% of the mean value).

The Ag-containing fibre series resulted in clearly higher (15–48%) TVOCs compared to the pure and rosin-containing PLA fibres at 25 °C. At 60 °C, the TVOCs were on the same level as for the fPLA10 series and 48% higher than for pure PLA fibres.

At a temperature of 105 °C, the chemically rich chemistry of rosin begins to disintegrate. This is obvious by noting that the melting temperature of rosin is ≈70 °C [6]. At 105 °C, the rosin-containing fibres resulted in 33–470% higher TVOCs compared to the reference fibres without rosin. The detailed (tabulated compounds) VOC reporting and chromatograms (total ion abundance as a function of retention time) per sample are included for readers as Appendix A. For the fPE10 series, some propanoic acid (2-methyl-) was observed and it was not indicated for the fPE fibres without rosin. Propanoic acid is naturally occurring and inhibits some moulds and bacterial strains [35]. For the fPLA10 series, terpineol (p-menth-1-en-8-ol) was clearly observed—this substance is used as biocide chemical and, likewise propanoic acid, it is due to the rosin in the fibres [36]. For PLA with rosin (fPLA10 series), also propanoic acid was observed, expectedly. In addition, limonene and pinene (monoterpenes) were found and these terpene molecules are typical in plant essential oils as well as in pine rosin [25,28,37]. A minor limonene trace was also found for the fPLA10 series. For many of the fibres, in general, caryophyllene and 1,4-methanoazulene compounds were observed—yet as minor traces for the fibres without rosin. These compounds are typical in plant essential oils and rosins [38] and presumably indicate that very slight remaining rosin could have been transferred within the preparation (spinning or sample collection) of rosin-containing and ‘pure’ fibres.

The fibres with Ag-doping resulted in rather high total emissions (TVOCs) at 105 °C. The chemical content was not ambiguous and reflected the typical PLA chemistry-related VOC release. In addition, a significant release of 2-pyrrolidinone was found contrary to pure PLA emissions. The 2-pyrrolidinone release stem from the poly(vinylpyrrolidone) surface treatment applied to the Ag particles [9]. It should be noted that PLA softens already at 55 °C [39], thus, the tested PLA polyfilaments have presumably begun to degrade at 105 °C.

### 4.3. Long-Term Performance of Polyfilament Fibres

The effect of UV exposure as a function of UV exposure time is presented in Figure 3a–c. It can be seen that the effect of UV ageing was essentially unaffected by the addition of rosin. For pure PE fibres as well as for the PE-rosin (10 w-%) fibres, most of the degradation occurred during the very first week of the irradiation period. This result is supported by the high values of strength (Table 5) for the rosin-modified fibres compared to the pure PE fibres. Rosin has been reported to be well integrated and dispersed to the polyethylene bulk in melt-spun fibres [6].

For the PLA fibres with and without rosin, the ageing trend was similar to PE fibres although the rosin-modified fibres were subject to a significant (absolute) impact and the ultimate strength was in practice lost during a total of two-week UV exposure. The previous work indicated lower quasi-static performance of rosin in PLA polyfilaments [6] compared to PE (and PA as well as polypropylene fibres). However, the low as-spun strength of the 0-exposure fibres (fPLA10 series), shown in Table 5, should be noted when comparing the pure and rosin-modified fibres in Figure 3. In addition, PLA degrades during the subsequent melting and processing cycles of rosin mixing. Moreover, rosin (its functional groups and terpenes) might have reacted with part of the migrants, i.e., lactic acid, lactoyllactic acid, even lactide. For this, the rosin-containing PLA fibres are not exactly comparable to the pure PLA fibres.

The results of the degradation of PE fibres, with and without rosin, due to standard washing suggest that rosin actually might have an observable negative effect on the washing durability. Noting that the initial, reference (0 washings) strength is 3.6% lower for the fPE10-160 series with rosin compared to pure PE fibres, the 24% increase in the degradation after ten washing cycles indicates a notable washing-related influence by rosin. This is expected because the melting and degradation of rosin itself begins at ≈70 °C [6]. For the melt-spun fibres with any local low-quality integration (i.e., clustering), the high temperature presumably leads to merging and diffusion of rosin inside the PE bulk. Contrary to all of the PE fibres, PLA fibres simply re-deform at 70 °C due to the low glass transition temperature ($T_{g}$ = 55 °C [39]) and hydrolysis at temperatures of 45–60 °C in hydro-thermal conditions [16]. After the temperature reaches the washing temperature of 70 °C in PLA, the filaments have already deformed and do not represent fibres and cannot be tested as filaments.

The results about the degradation related to UV radiation and washing revealed significant degradation of fibres in realistic application-simulating environments. For better understanding and development of the fibre chemistry, it is important to know how great is the pure thermal ageing—without the effects of UV and moisture. Figure 4 shows the results of thermal ageing at 60 °C. The effects of the long-term elevated temperature on the PE fibres with and without rosin are interesting—the effects are dependent on the melt-spinning temperature (sensor of the spinneret unit). The effect of the processing temperature on fibre strength (tenacity) has been reported earlier [6] and is evident from the as-spun strength values in Table 5. It is important to note that the degrading effect of long-term elevated temperature and the observed dependence on the spinning temperature were not observed to be dependent on the addition of rosin. This means that the degradation of rosin does not begin when the temperature is below 70 °C—even when the conditioning is continued for several weeks. The PLA fibres naturally lost their fibrous form at 60 °C during the very first measuring interval (seven days) and could not be tested.

Along with mechanical performance, the antibacterial response should remain in the polyfilaments during anticipated operation. The durability of the antibacterial response against *S. aureus* by rosin in cellulose acetate nanofibre networks is reported in the current literature—for as long as a two-month storage period at room temperature [34]. In this research, it was confirmed that the response against *S. aureus* remained after the thermal ageing period of PE-rosin polyfilaments (four weeks at 60 °C). As shown in Figure 5, the bacterial content (*S. aureus*) in the medium with fibres (fPE10-180 series) decreased from $10^{5}$ cfu/mL in 24 h to below the detection limit (3 × $10^{1}$ cfu/mL).

## 5. Discussion

The results of mechanical testing for the fibre durability showed that PE fibres with and without rosin are durable and strong even when relatively large rosin concentration (10 w-%) is used during the polymer blending. The significant effect of the spinning temperature on the long-term thermal ageing (60 °C) emphasized the essence of processing-related parameters and induced phenomena, such as crystallinity [40,41] and for some events cross-linking [42], when using natural components, like rosin. The processing parameters are typically adjusted per specific application and design criteria of the polyfilament. However, in this study, the strength deterioration or development was not found observably dependent on the addition of rosin. The change in the measured ultimate strength in time was similar by its trend for pure and the rosin-modified fibres. The reported thermal gravimetric data [6] for the exact rosin and the VOC analysis of this study suggest that some amount of rosin (components) is emitted or degraded from the polymer system for temperatures above 60 °C. Based on the VOC data, terpene and acid functional groups in general—that are typically reported to be responsible for the antibacterial response—are at least partly removed. The removal of these compounds did not finally affect the antibacterial response (at the analysed bacterial input level and mixing time of 24 h), even after four weeks of thermal ageing at 60 °C. Therefore, it is suggested that the relatively high rosin content applied to the polyfilaments in this study let the antibacterial response to remain despite the immediate changes (by emissions).

The melt-spinning of PLA fibres with natural antibacterial response and with adequate mechanical durability as the main targets was noticed challenging in this study. The results about the poor performance at 60 °C (lost integrity) are strongly supported by the current literature [43]. For a cellulose-based filler, Aouat et al. reported that it was not possible to even determine Young’s modulus after nine days of conditioning at 60 °C independent of the dosage of natural additives (cellulose) [16]. In summary, when noting the results of antibacterial response of the fibres modified by the selected Ag-containing additive, the selected PLA (grade) is less preferred polymer basis for polyfilament melt-spinning with rosin compared to the developed PE compounds with rosin.

## 6. Conclusions

The long-term durability and ageing due to the anticipated environment of application (i.e., textiles, industrial filters) of polyfilaments were studied for PE and PLA melt-spun polyfilaments with pine gum rosin as natural antibacterial additive. The analysis of weathering in this study included a UV irradiation period of 4–5 weeks, standard washing at 70 °C, and thermal ageing at 60 °C for four weeks. The release of volatile organic compounds was studied using the standard VOC analysis at temperatures of 25 °C, 60 °C and 105 °C. The antibacterial response against Gram-positive *S. aureus*, as reported earlier for several rosin-containing polymer fibres, was compared with fibres modified by rival synthetic Ag-containing additive here. In the end, the antibacterial response of the mechanically durable PE fibres with and without rosin was confirmed—after four weeks of thermal ageing. The results revealed the following outcomes:A high 20 w-% of rosin content was be applied for melt-spun PE and PLA polyfilaments by using a copolymer-type surfactant (F-127, Pluronic) but the ultimate mechanical properties at ambient conditions were not improved;The total VOC emissions from the melt-spun PE fibres were lower for those fibres that were modified with 10 w-% of rosin and when analysed over a temperature range of 25…60 °C;PE fibres with a 10 w-% of rosin were mechanically durable against UV-radiation, thermal ageing at 60 °C and standard washing cycles at 70 °C as well as in terms of a strong antibacterial response against *S. aureus*.

## Figures and Tables

**Figure 1 molecules-26-00876-f001:**
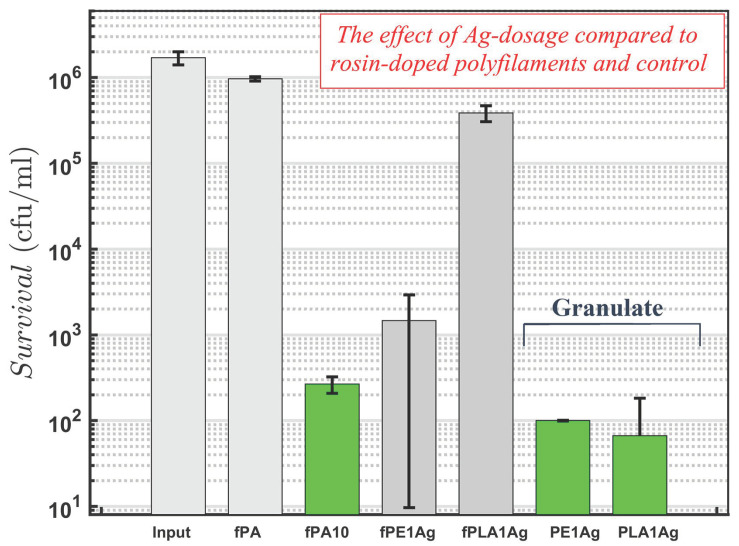
Antibacterial response of melt-spun polyfilmants against *S. aureus* in liquid (Ringer’s solution, 1/4 strength) after 24 h mixing at room temperature: Ag-containing PE and PLA samples with 1 w-% of additive concentration in both fibrous (polyfilament) and granulate form (prior to melt-spinning) are compared to polyamide-6 [6] control and reference samples (fPA, fPA10). The detection limit is 3 × $10^{1}$ cfu/mL.

**Figure 2 molecules-26-00876-f002:**
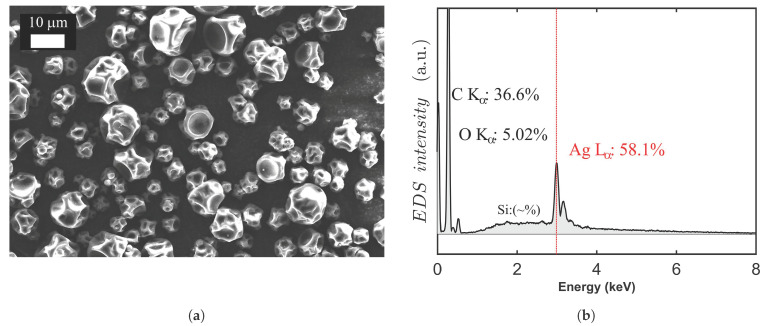
Characteristics of the Ag nanoparticles: (**a**) SEM imaging of particle size variation and particle morphology; (**b**) the EDS spectrum measured on Ag nanoparticles and the elemental distribution.

**Figure 3 molecules-26-00876-f003:**
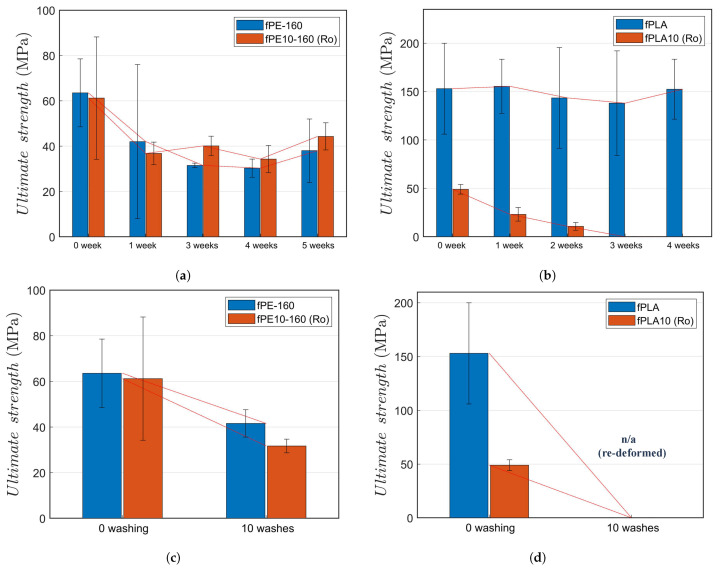
The effect of UV exposure on the ultimate strength of: (**a**) PE fibres with and without rosin; (**b**) PLA fibres with and without rosin. The effect of standard washing (70 °C) on the ultimate strength of: (**c**) PE fibres with and without rosin; (**d**) PLA fibres with and without rosin.

**Figure 4 molecules-26-00876-f004:**
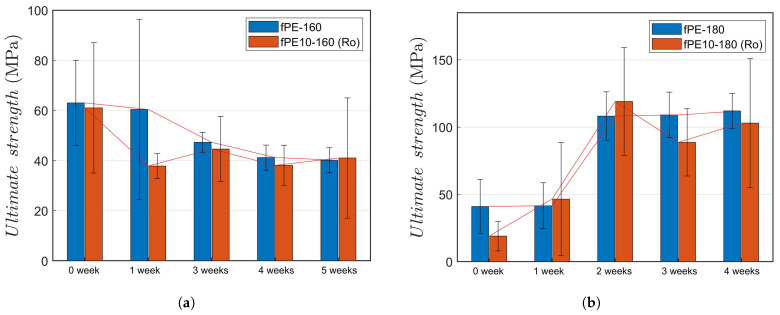
The effects of thermal ageing at 60 °C: (**a**) PE fibres (with melt-spinning temperature of 160 °C) with and without rosin; (**b**) PE fibres (with melt-spinning temperature of 180 °C) with and without rosin. The PLA series fibres lost their fibrous form and could not be tested.

**Figure 5 molecules-26-00876-f005:**
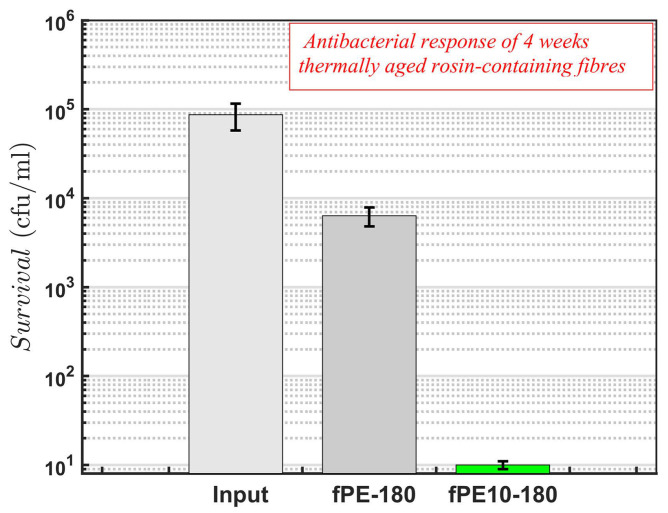
Antibacterial response of melt-spun polyfilaments against *S. aureus* in liquid (Ringer’s solution, 1/4 strength) after 24 h mixing at room temperature: four weeks thermally aged (60 °C) PE polyfilament fibres with and without rosin. PLA fibres lose their fibrous form at 60 °C and can not be tested as polyfilament fibres. The detection limit is 3 × $10^{1}$ cfu/mL.

**Table 1 molecules-26-00876-t001:** Polymers for preparing antibacterial multifilament fibres.

Polymer Basis	Grade	Provider
High-density polyethylene (HDPE)	CG9620	Borealis Polymers
Poly lactic acid	2003D	Ingeo/NatureWorks

**Table 2 molecules-26-00876-t002:** Processing parameters for melt-spinning of rosin-containing blends.

Polymer	Rosin % (*w*/*w*)	Temperature (°C)	Series Names
PE	0, 10	160, 180	fPE-160, fPE10-160, fPE10-180
PLA	0, 10	160–180	fPLA, fPLA10

**Table 3 molecules-26-00876-t003:** Processing parameters for melt-spinning of surfactant (PF) containing blends.

Polymer	PF/Ro, Ro	Temperature (°C)	Series Names
PE	0.01, 20% (*w*/*w*)	160	fPE1PF
PLA	0.01, 20% (*w*/*w*)	160	fPLA1PF

**Table 4 molecules-26-00876-t004:** Processing parameters for melt-spinning of silver additive (Ag) containing blends.

Polymer	Ag % (*w*/*w*)	Temperature (°C)	Series Names
PE	1	180	fPE1Ag
PLA	1	180	fPLA1Ag

**Table 5 molecules-26-00876-t005:** Mechanical properties of fibres and the determined mechanical properties per additive concentration, respectively.

Fibre Series	Additive(s)	Ultimate Strength (MPa), Respectively	Strain (δL/*L*) at Break (%), Respectively
fPE-160, from [6]	Ro (0, 10 w-%f)	63 ± 17, 61 ± 26	1130 ± 95, 1528 ± 210
fPE-180, from [6]	Ro (0, 10 w-%)	41 ± 20, 19 ± 11	2076 ± 242, 1118 ± 225
fPE1PF	PF, Ro (20 w-%)	38 ± 16	1234 ± 491
fPE1Ag	Ag (1 w-%)	53 ± 47	1407 ± 574
fPLA, from [6]	Ro (0, 10 w-%)	153 ± 47, 49 ± 5	294 ± 60, 3 ± 0
fPLA1PF	PF, Ro (20 w-%)	77 ± 30	4 ± 1
fPLA1Ag	Ag (1 w-%)	146 ± 47	231 ± 73

**Table 6 molecules-26-00876-t006:** Average and range (deviation) of total volatile organic compound (TVOC) (ng/L/g) measured from the parallel polyfilament samples at different temperatures. Collection between 6.0 and 36.4 min (retention time) per sample and temperature. Chromatograms are given in the Appendix A.

Fibre Series	Temperature 25 °C	Temperature 60 °C	Temperature 105 °C
fPE	381 (255–308)	8386 (6252–10,520)	160,092 (137,863–182,321)
fPE10	113 (113–114)	6637 (6434–6840)	214,205 (200,338–228,071)
fPLA	40 (39–41)	766 (532–1000)	19,382 (18,531–20,233)
fPLA10	31 (0–31)	1120 (691–1548)	929,724 (568,861–1,290,588)
fPLA1Ag	46 (30–61)	1132 (442–1832)	38,693 (33,419–43,967)

## Data Availability

Data available by request until 2023.

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
