# Peer review of "Weathering of Antibacterial Melt-Spun Polyfilaments Modified by Pine Rosin"

_molecules, 2021, doi:10.3390/molecules26040876_

Round 1

Reviewer 1 Report

The manuscript Weathering and safety of antibacterial polymer-rosin polyfilaments by Kenerva et al. is quite interesting. However some points required more clarification by the authors 

Among these points:

1- The title is highly misleading since the safety is usually related to toxicological studies which is totally absent in the work 

2- The abstract should be restructured to give more digital results rather than elastic sentences

3- The VOC analysis is poorly presented; the authors should submit the chromatograms as supplementary and should present the results in a better table to give more details rather than that presented in table 6 

4- The authors checked the effect of different weathering factors on the shape of the particles which was presented by the SEM photos but the effect may be not related to the shape of the particles it might be to the size of the particles

Author Response

Response attached as PDF

Reviewer 2 Report

The authors prepared various types of specimens and investigated their mechanical properties, weathering and safety. Regrettably, manuscript is not well-structurized and inconsistent specimen selection for each figures and tables make the manuscript difficult to understand. Although there many types of specimens were prepared, the results are shown for only limited ones. The authors are recommended to select important ones to clarify the points that the authors would like to show in the present study. Please show the results as figures and tables for fixed selected specimens. It is regrettable the manuscript cannot be judged in the present form. The authors are encouraged to revise the manuscript according to the following comments and resubmit it as a new one after additional experiments.

(1) Too many samples (Tables 1-4) were prepared, but the results are shown for only limited ones as in Figs. 2-4 and Tables 5 and 6 in the inconsistent way. Please select important ones to clarify the points that the authors would like to show.

(2) Inconsistent way of showing data in figs and tables. Fig. 2 shows the data for PE, PA, PLA but those for PA and PLA are lacking in Fig. 3. Also, data for the specimens with Ag or GO are lacking in Figs 2 and 3. The specimens used should be same for all figures and tables.

(3) Figure 4 should show the data for neat PE, PLA, PA and those with rosin or Ag.

(4) The statements why the authors used GO and Ag as additives should be included in the Introduction section not in other sections.

(5) The reason why the authors selected the specific concentrations for rosin, GO, and Ag should be stated.

(6) Specimens with 20 wt% Rosin were prepared for PE but not for PA and PLLA (Table 2). Why it that ? The same remark for specimens with 2% Ag only for PE not for PLA.

(7) Average molecular weight and polydispersity index are lacking for PE, PLA, and PA. The type of PE (HDPE, LDPE, or LLDPE), D-lactic acid unit concentration of PLA, and monomer units of PA (or actual polymer name) are not specified.

(8) For Tables 5 and 6, please do not show all the data the authors took but show the results for important specimens as figures.

(9) GO/Ro of the second specimen in Table 4 should be 0 or something should be wrong.

(10) What are the granulate specimens in Figure 4? Please specify the preparation procedure. Why do they appear suddenly here? For consistency, the data for the granulate specimens should be deleted.

Author Response

Response attached as PDF.

Reviewer 3 Report

M. Kanerva and coauthors reported the weathering and antibacterial property of polymer-rosin polyfilaments. At least for me, the selling point of this manuscript and the scientific significance of this work are not very clear. The experiment design is loss logic. Furthermore, lots of claims and hypotheses in the manuscript are absent from the supporting of experiment results. Lots of results are not well discussed and explained. For example, there is no evidence to indicate whether the decrease of fiber strength results from the degradation of the polymers or the degradation and/or leach out the additives (e.g., the rosin.) under UV irradiation and thermal treatment. Besides, improvement in English is also required. I don’t think it is a meaningful research work ready for publication at the current version. I would like to suggest the authors rearrange the structure of the manuscript, rethink the significance of this work, and redesign some of the experiments before considering publication.  

Author Response

Response attached as PDF.

Reviewer 4 Report

From a microbiological point of view, the work presents the important limitation; only one strain of S. aureus is used to analyse this characteristic.
It is necessary to extend the study to include different bacterial and fungal species

Author Response

The response to Reviewer 4 is given in the attached PDF:

Reviewer 5 Report

The research proposed by Mikko Kanerva and coll., focused on the usage of pine rosin as natural antibacterial chemical and analyzed the weathering of melt-spun polyethylene  (PE) and poly lactic acid (PLA) polyfilaments.

The reviewer has some questions:

What applications are envisaged for this study? In which field these results could be interesting to be applied? This should be clearly specified in the article.

In Figure 1 the error bars are not “standard”. What they represent? How were calculated?

Author Response

The response to Reviewer 5 are attached as PDF:

Round 2

Reviewer 1 Report

The manuscript has much improved and the authors responded with most of the comments

Author Response

Our team appreciate the comments Reviewer I sent during the first review round and response to our changes. The manuscript was also improved based on comments of other 4 reviewers.

Reviewer 2 Report

Some issues were improved but many crucial issues remained unimproved.

(1) The paper mainly focuses on the effects of rosin as the authors say in the Background section. However, Figure 1 shows mainly the effects of Ag and the effect of rosin is limited to PA. At least all the figures and tables should have the data for PA, PE, PLA with and without rosin. If the authors do not have sufficient data for PA with and without rosin, all PA data are recommended to be removed. Also, Figure 2 shows the characterization of Ag. There is no need for Figure 2.

(2) The crucial issues of inconsistent way of showing data and the lacking of blank samples remain. For examples, Figure 1 should show the data of fPAAg, fPE, fPLA, fPE10 and fPLA10.

(3) Considering the rosin concentrations of 0 and 10% in Table 2, the rosin concentration in samples with PF in Table 3 should be 10%. With rosin concentration of 20% in the samples with PF, there is no blank samples at the same rosin concentration (10%) without PF.

(4) Average molecular weight and polydispersity index are lacking for PE, PLA, and PA. The type of PE (HDPE, LDPE, or LLDPE), D-lactic acid unit concentration of PLA, and monomer units of PA (or actual polymer name) are not specified.

(5) PA data suddenly appears in Figure 1 without any statements in the Experimental section.

(6) Tables 1-3 can be unified into one table.

(7)Table number starts from Table 5.

(8) f in the sample abbreviations does not seem to be required.

(9) Background section should be included in Introduction section.

Author Response

We have made a written complaint to Molecules editorial board about Reviewer's (2) comments that are not professional and merely indicate that Reviewer 2 did not even read our manuscript - not the first version and not the revised version - and made up unprofessional comments. Our team and the journal editors no longer wait response from Reviewer 2.